# *Ustilago maydis* Metabolic Characterization and Growth Quantification with a Genome-Scale Metabolic Model

**DOI:** 10.3390/jof8050524

**Published:** 2022-05-20

**Authors:** Ulf W. Liebal, Lena Ullmann, Christian Lieven, Philipp Kohl, Daniel Wibberg, Thiemo Zambanini, Lars M. Blank

**Affiliations:** 1iAMB-Institute of Applied Microbiology, ABBt, RWTH Aachen University, Worringerweg 1, 52074 Aachen, Germany; lena.ullmann@rwth-aachen.de (L.U.); philipp.kohl@rwth-aachen.de (P.K.); tiemo.zambanini@rwth-aachen.de (T.Z.); 2Unseen Biometrics ApS, DK-2800 Kgs. Lyngby, Denmark; christian.lieven@gmx.de; 3Novo Nordisk Foundation Center for Biosustainability, Technical University of Denmark, DK-2800 Kgs. Lyngby, Denmark; 4Genome Research of Industrial Microorganisms, CeBiTec, Bielefeld University, 33501 Bielefeld, Germany; dwibberg@cebitec.uni-bielefeld.de

**Keywords:** *Ustilago maydis*, genome-scale metabolic model, constraint-based model, biotechnology, COBRA, FBA, metabolism, itaconate

## Abstract

*Ustilago maydis* is an important plant pathogen that causes corn smut disease and serves as an effective biotechnological production host. The lack of a comprehensive metabolic overview hinders a full understanding of the organism’s environmental adaptation and a full use of its metabolic potential. Here, we report the first genome-scale metabolic model (GSMM) of *Ustilago maydis* (iUma22) for the simulation of metabolic activities. iUma22 was reconstructed from sequencing and annotation using PathwayTools, and the biomass equation was derived from literature values and from the codon composition. The final model contains over 25% annotated genes (6909) in the sequenced genome. Substrate utilization was corrected by BIOLOG phenotype arrays, and exponential batch cultivations were used to test growth predictions. The growth data revealed a decrease in glucose uptake rate with rising glucose concentration. A pangenome of four different *U. maydis* strains highlighted missing metabolic pathways in iUma22. The new model allows for studies of metabolic adaptations to different environmental niches as well as for biotechnological applications.

## 1. Introduction

*Ustilago maydis* is a model organism and economically important fungus from the division of *Basidiomycota*. The associated corn smut disease affects maize harvest, but the tumors are also used as food [1]. As a parasite, *U. maydis* grows into the plant tissue to extract substrates for its own metabolic activity. *Ustilaginaceae* show a versatile product spectrum, such as organic acids (e.g., itaconate, malate, succinate), polyols (e.g., erythritol, mannitol), and extracellular glycolipids, which are considered value-added chemicals with potential applications in the pharmaceutical, food, and chemical industries. *U. maydis* has developed an effective native production of itaconic acid, an important platform chemical. Indeed, the itaconic acid production in *U. maydis* was improved to surpass the current biotechnological route of *Aspergillus terreus*-based production. The advantages are yeast-like growth; high productivities, yields, and titer; and reduced byproduct formation [2,3,4], and since it is a model organism, efficient genetic tools are available [5].

The annotated genome sequence of *Ustilago maydis* strain 512 enabled a deeper understanding of its pathogenic mechanisms as well as metabolic competencies [6]. Annotated genomes can be used to construct genome-scale metabolic models (GSMM), which serve as a knowledgebase of metabolic capacities and allow rational biotechnological engineering [7]. GSMM can be optimized to identify genetic modifications for metabolic engineering that maximize the production of metabolic intermediates. In addition, optimal biotechnological production routes regarding different organisms and metabolic pathways can be computationally evaluated using their respective GSMM [8]. The performance of metabolic microbial and cross-kingdom interactions can be interrogated in order to identify exchange metabolites, community stability, and metabolic properties that mark transitions from mutualism to parasitism [9,10,11].

Here, we present the first high-quality genome-scale metabolic model for *U. maydis*, called iUma22. Growth phenotype assays based on BIOLOG with 190 substrates were conducted to reveal the metabolic versatility of *U. maydis* for more realistic model predictions in native habitats. Moreover, growth kinetics across a range of high glucose concentrations were performed, which allowed for improved metabolic characterization during biotechnological fermentations. To judge the metabolic completeness of iUma22, as well as metabolic capacities in comparison to the *Ustilago* genus, a pangenome of annotated enzymes of different *U. maydis* strains was constructed. The model is freely available online via the Biomodels database ID: MODEL2203250001 and GitHub: https://github.com/iAMB-RWTH-Aachen/Ustilago_maydis-GEM (accessed on 24 March 2022). The quality of the model was assessed using the Memote evaluation tool, as well as FROG reports for reproducibility (see Appendix A). 

## 2. Materials and Methods

### 2.1. Draft GSMM from Pathway Tools

The genomic DNA sequence of *Ustilago maydis* (Strain 521 FGSC 9021) [6] was obtained from NCBI’s RefSeq project [12]. A corresponding annotation file was then exported from the MIPS Ustilago Maydis Database via the PEDANT Interface [13]. Using the PathoLogic Tool [14], the sequence and annotation files were parsed and, in combination with the MetaCyc reactions database, a new Pathway/Genome Database (PGDB) was created. During pathway cleaning, reactions from other taxa are pruned, unless there were enzymes matching all of the reactions. Additional metabolic activity was identified using the ‘Pathway Hole Filler’ function and the sequence information of isoenzymes was used to query the proteome of *U. maydis* via pBlast. Protein sequences were queried on PEDANT, MUMDB, MetaCyc or KEGG [15,16] and manually curated, while inconclusive polypeptides, as well as those that are involved in signaling and other nonmetabolic pathways, were discarded. While PEDANT and MUMDB are discontinued, information on the sequence and annotation for *U. maydis* can be accessed via EnsemblFungi: https://fungi.ensembl.org/Ustilago_maydis/Info/Index (accessed 24 March 2022) [17], MycoCosm: https://mycocosm.jgi.doe.gov/Ustma2_2/Ustma2_2.home.html (accessed on 24 March 2022), NCBI (genome assembly ID: 225285), and Uniprot (Proteome ID: UP000000561).

### 2.2. Strains Sequenced, Pangenome, KEGG Pathway Enrichment 

To identify metabolic differences within the *U. maydis* strain family, a pangenome consisting of five *Ustilago maydis* strains was assembled, including strains 198, 482, 485, and 512 [18]. The Nanopore Rapid DNA Sequencing kit (SQK-RAD04, Oxford Nanopore Technologies, Oxford, UK) was used for preparation, and sequencing was performed on an Oxford Nanopore GridION Mk1 sequencer using a R9.4.1 flow cell. The Nextera XT DNA Sample Preparation Kit (Illumina, San Diego, CA, USA) was used for whole-genome-shotgun PCR-free libraries from 5 μg of gDNA. The library quality was assessed by an Agilent 2000 Bioanalyzer with Agilent High Sensitivity DNA Kit (Agilent Technologies, Santa Clara, CA, USA) for fragment sizes of 500–1000 bp. Paired-end sequencing was performed on the Illumina MiSeq platform (2 × 300 bp, v3 chemistry). Adapters and low-quality reads were removed by an in-house software pipeline prior to polishing, as recently described in [19]. Run control was based on MinKNOW (Oxford Nanopore Technologies, Oxford, UK) with the 48-h sequencing run protocol. Base-calling was performed offline using a Bonito assembly with canu v2.1.1 [20], contigs were polished with Pilon [21] for ten iterative cycles, and for read mapping, BWA-MEM [22] and Bowtie2 v2.3.2 [23] were used in the first and second five iterations, respectively.

Genes were predicted using GeneMark-ES 4.6.2. [24] and functionally annotated using a modified version of the genome annotation platform GenDB 2.0 [25] for eukaryotic genomes [26]. Similarity searches were conducted against COG [27], KEGG [16] and SWISS-PROT [28]. Identification of putative tRNA genes was conducted with tRNAscan-SE [29]. Completeness, contamination, and strain heterogeneity were estimated with BUSCO (v3.0.2 [30]) using the fungi-specific single-copy marker genes database (odb9). The obtained genome sequences are compared and documented in more detail in Ullmann et al. (2022) [18]. The pangenome of all available *U. maydis* strains was calculated by means of EDGAR 3.0 [31]. The KEGG pathway annotation was performed by comparison of E.C. numbers in the pangenome annotation and E.C. numbers in the reaction description of iUma22. The comparison resulted in three lists: E.C. numbers only present in iUma22 (iUmaNOTpan), present in the pangenome and iUma (iUmaANDpan), and only present in pangenome (panNOTiUma). The panNOTiUma list was exported as a fasta-file, and KAAS [32] was used to annotate the list with KEGG pathway information. The annotation of the genes in the SBML-file was achieved with the BioServices Python package [33]. 

### 2.3. Biomass Equation and Growth/Non-Growth Maintenance 

The composition of proteins, RNA, and DNA was estimated based on the respective protein and genome sequences, whereas the composition of lipids and the cell wall were results-mined from scientific articles [34,35]. The exact biomass composition of *U. maydis* is not available; however, the specific elemental composition [36] (and the biomass composition for fungi in general [37]) was used as a starting point, and linear programming was applied to approximate the total biomass composition (Appendix A). The composition values of each monomer were converted into stoichiometric values [38]. For example, to determine the amino acid (AA) composition contribution (in mol_AA_/g_Prot_), first, the AA-protein molarity (MP_AA_ in g_AA_/mol_Prot_) was calculated by multiplying the AA codon frequency with the AA molar mass (minus the molar mass of water released during polymerization) (Equation (1)). Normalizing each AA-protein molarity by the overall sum yields the weight fraction of each AA (WP_AA_) (Equation (2)). Division of the AA weight fraction (WP_AA_) by its molar mass and multiplication with the weight fraction of protein to the dry weight (*X* in g_Prot_/g_CDW_) and conversion from mol to mmol (factor 1000) provides the stoichiometric factor (SF_AA_) (Equation (3)). To calculate the stoichiometric factor of an AA (*SF_AA_*), the molar percentage (*MP*) is multiplied with the fractional protein mass per biomass (*X*) (Equation (2)).
MP_AA_ [g_AA_/mol_Prot_] = CDN_AA_/Σ CDN • (M_AA_ − M_H2O_)(1)
WP_AA_ [g_AA_/g_Prot_] = MP_AA_/Σ MP(2)
SF_AA_ [mol_AA_/g_CDW_] = WP_AA_/(M_AA_ − M_H2O_) • *X* • 1000(3)

The fraction of protein on total biomass (*X*) is unknown and was determined by linear optimization. The average elemental composition of each macromolecule (protein, DNA., RNA, lipid, cell wall) was determined by summing up the products of the absolute amount of each element, and the corresponding C-mole content was calculated [36]. Phosphorous and Sulfur were added from the elemental composition of *Sacharomyces cerevisiae*. The optimization followed the formula:A • *X* = b(4)
subject to: x_lbi_ ≤ x_i_ ≤ x_ubi_(5)

The rows of matrix A correspond to elements *C*, *H*, *O*, *N*, *P*, and *S*, whereas the columns correspond to the macromolecule types (protein, DNA, RNA, lipids, cell wall). Vector b represents the measured elemental biomass, *C*, *H*, *O*, and *N*, supplemented by the elemental content of the *P* and *S* of *S. cerevisiae*. The equation was solved for vector *X*, the biomass fractions of each of the macromolecules (Table 1 and Appendix A). The ATP-associated growth-maintenance (GAM) with ATP-dependent glucose uptake was calculated as 31 mmol/g_CDW_ by optimization of the sum of the squared errors of growth experiments (see Section 3). The nongrowth-associated maintenance (NGAM) was calculated as 0.75 mmol/g_CDW_/h by maximization of NGAM with the lowest, not infeasible glucose uptake rate of 0.22 mmol/g_CDW_/h close to the experimentally determined glucose maintenance requirement of 0.2 mmol/g_CDW_/h (see Section 3.3).

### 2.4. Substrate and Growth Experiments

For the substrate utilization experiments, the BIOLOG Phenotype Microplates™ PM1 and PM2A were used with *Ustilago maydis* strain FB1. Cultures were first grown on YEPS-agar plates at 30 °C for at least 24 h. To prepare the precultures, 25 mL of YEPS medium were inoculated from the plates of each strain then performed in 100 mL Erlenmeyer flasks and incubated at 30 °C, 200 rpm for 24 h (Ecotron Incubation shaker, Infors HT AG, Bottmingen, Switzerland). The inoculation fluid was prepared with IFY-0 (1.2×), cell suspension, and sterile water to obtain a starting turbidity of 62% T, with 100 μL for each well. The inoculated plates were shaken at 200 rpm with a shaking diameter of 50 mm at 30 °C and with a humidity of 70% for up to 168 h (Multitron Incubation shaker, Infors HT AG, Bottmingen, Switzerland). Microbial growth was measured with the SynergyMX (BioTek Instruments, Winooski, VT, USA) with an optical density of 600 nm. The BIOLOG raw data is available in the Appendix A.

The threshold for positive growth was determined by examining the optical density (OD) histograms for each plate. A normal distribution at low OD values represents the OD range below positive growth (see Section 3.2). The final growth threshold of 0.4 a.u. (absorbance units) was empirically determined to maximize logic consistency and to minimize the integration of false positive metabolic activity. The value approximates the end of a normal distribution of non-growth at low ODs. Separate glucose shake-flask batch experiments in modified Tabuchi medium [39] were conducted with strains MB215, FB1 mating type a1b1, and literature data was used to estimate growth rates and glucose uptake rates. The OD measurements of the growth data were converted into g_CDW_/L using the empirical relation from yeast of 0.62 g_CDW_/L /OD (BNID 111182, [40]). The growth rates were identified using a nonlinear fit of the biomass to the Verhulst equation,
X(t) = X_0_*C/(X_0_ + (C − X_0_)*exp(−μ*t)),(6)

Which calculates the biomass from the initial biomass (X_0_), the max biomass capacity (C), the growth rate (m), and time (t). The substrate uptake rate was estimated with a linear equation [41].

## 3. Results and Discussion

### 3.1. Description of iUma22

The genome-scale model for *Ustilago maydis* was constructed based on the genome sequence and annotation of strain 521 [6]. Table 2 shows the number of represented genes, metabolites, and reactions in the new reconstruction, as well as a comparison to the community yeast model ([42], version 8.5.0). Whereas the community yeast model was more comprehensive, iUma22 had a higher gene-to-reaction ratio as well as gene-protein-reaction relationships (GPR), as we aimed to include well-connected metabolic pathways with (predicted) annotated genes. *U. maydis* and *S. cerevisiae* have a similar number of predicted genes and, when assuming yeast 8.5.0 as a benchmark of metabolic representation, the iUma22 reached 70% of genes completeness. There are likely gaps in the secondary metabolism discussed in Ullmann et al. (2022) [18], as well as adaptations to the pathogenic lifestyle.

The quality of iUma22 was tested with Memote with an overall performance of 57% [43] (Figure 1). Mass and charge balance, as well as metabolite connectivity, showed high quality, with scores of over 98%. Memote detected unbounded fluxes that reached boundary conditions during flux variability analysis for 203 reactions on standard media. The stoichiometric consistency of the model could not be evaluated, thus decreasing the overall consistency quality to 53%. Note, however, that for the *S. cerevisiae* community model ([42], version 8.5.0), the stoichiometric consistency test also failed. Annotations for metabolites, reactions, and genes contained detailed and unique annotations. The community yeast model, developed for more than a decade, was evaluated by Memote with a total score of 65%.

### 3.2. Carbon Substrate Tests with BIOLOG Phenotype Arrays

The model iUma22 correctly reproduced 96% of the growth phenotypes tested in BIOLOG carbon source assays. In these assays, each well of a 96-well plate was equipped with a different substrate; we chose carbon substrate plates PM1 and PM2A from the manufacturer (an overview of the carbon source distribution in the wells is provided in the Appendix A). Substrate utilization was tested photometrically, and we chose an OD threshold of 0.4 a.u. for growth, a value right after the apparent normal distribution of nongrowth for low final OD values (Figure 2). The threshold was a compromise to include growth for glycine dipeptides (PM1: E1, G1, G6, H1), but also included TCA cycle intermediates (succinate (PM1:A5), fumarate (PM1:F5), aspartate (PM1:A7), and malate (PM1:G12)). These TCA cycle intermediates could not be enabled for growth in the model. The largest set of reactions added because of the BIOLOG plates included di- and oligosaccharide metabolism and methylated central carbon metabolites (detailed list in the Appendix A). Overall, growth took place in 52 wells (36 in PM1 and 28 in PM2A). iUma22 was manually adjusted to reproduce the majority of the growth phenotypes (Figure 2).

While the majority of the substrates were correctly reproduced, some metabolites failed to support growth in iUma22. Many intermediates from the TCA cycle did not support growth in the BIOLOG results (2-oxoglutarate, fumarate, succinate, aspartate) while others, such as lactate (PM1:B9), malate (PM1:G12), and succinamate (PM2A:F10), were positive (Figure 2). The degradation of arginine (PM2A:G4), isoleucine (PM2A:G9), and ornithine (H1) was dysfunctional in iUma22, although proline had been predicted to enable growth. These metabolites are carbon intermediates of the common carbon metabolization following the ornithine-glutamate aminotransferase reaction (E.C. 2.6.1.13, ORNTArm). The metabolization of sebacic acid (PM2A:F8) is not represented in the model, because no information is available in the databases KEGG and Metacyc. False positive growth predictions indicated exchange and transport reactions of metabolites in the model that could not actually be imported. Thus, growth predictions of the corresponding metabolites were corrected by removing exchange reactions of the associated metabolites.

### 3.3. Growth Rate Correlation

We evaluated glucose growth characteristics and compared the results with iUma22 predictions. We used published batch data of *U. maydis* MB215 control strains [4] that were genetically modified for optimized biotechnological performance, as well as newly generated data (Table 3). Note that the model considers ATP-consuming glucose uptake via Hxt1 [44]. Experiments with high initial glucose concentrations of >50 g/L resulted in lower glucose uptake rates, which is not associated with osmotic stress, because the yield stayed high with increasing glucose levels. The correlation between substrate uptake rate and growth rate (Figure 3A) was strong, R^2^ = 0.99, with a realistic growth-associated yield (slope) of 0.47 g_CDW_/g_glc_. While experiments ‘130v1’ and ‘130v2’ displayed disparate substrate uptake and growth rates despite similar initial glucose concentrations, their yield was comparable, indicating a similar metabolic state. The maintenance parameter of the high-yield experiments was calculated as the x-axis interception, i.e., glucose uptake in the absence of growth, and resulted in 0.2 mmol/g_CDW_/h, comparable to *S. cerevisiae* (0.2 mmol/g_CDW_/h [45]). The growth predictions of iUma22 are closely captured in the experimental results (Figure 3C).

### 3.4. U. maydis Pangenome Comparison

We used the available sequenced *U. maydis* strains [18] to compare the enzymatic gene inventory among strains and regarding iUma22. A pangenome of strains 198, 482, 485, and 512 was constructed by means of EDGAR 3.0, resulting in 7838 coding genes, of which 1458 are annotated with an E.C. number. We explored how many genes are shared among all strains and used strain 512 as a reference to identify unique enzyme coding genes and the proportion of shared genes to other strains (Figure 4A). An overwhelming number of genes is shared among all strains (‘all strains’ in Figure 4A). Strain 512 has a number of unique enzyme-coding genes that are more likely shared with other strains because genes with more selective distributions (shared among three strains to one other strain) are becoming less frequent (see also Ullmann et al. [18], 2022). We then evaluated how the genetic composition of iUma22 differed with respect to the *U. maydis* pangenome of E.C.-annotated genes (Figure 4B). Table 4 shows the top five pathways with the most enzyme annotations for iUma22-unique genes, shared genes, and pangenome-unique genes with E.C. numbers. The majority of iUma22-unique genes belong to oxidative phosphorylation, and the unique genes in the strain pangenome belong to diverse central carbon metabolic pathways (Table 4). Particularly noteworthy is the inositol phosphate pathway (Figure 4C), not only because of the highest number of pangenome unique metabolic capacity but also because inositol was a growth-supporting substrate of the BIOLOG, which was manually added to the model.

## 4. Conclusions

Here, we present iUma22, a genome-scale metabolic model of *U. maydis,* which correctly simulates a large number of substrate phenotypes as well as glucose-based growth rates. The decrease in glucose uptake at high concentrations indicates potential for biotechnological optimization. The model can be further used to identify the biotechnological potential of metabolite overproduction and to optimize metabolic engineering strategies. It can also be used to study metabolic shifts in different life cycles of the fungus during plant infection. While the reconstruction was performed based on model strain 521, the genome sequencing of additional *U. maydis* strains provided insight into additional metabolic pathways, which could be used to generate a pangenome-scale metabolic model of *U. maydis*.

## Figures and Tables

**Figure 1 jof-08-00524-f001:**
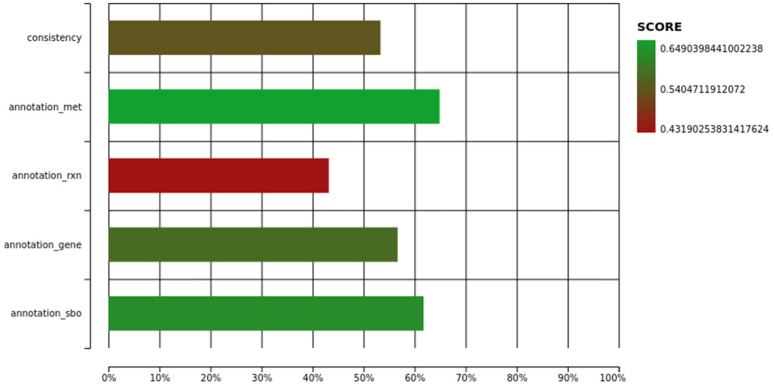
Memote quality report of iUma22 with total score of 57%. The full HTML report is provided as Appendix A.

**Figure 2 jof-08-00524-f002:**
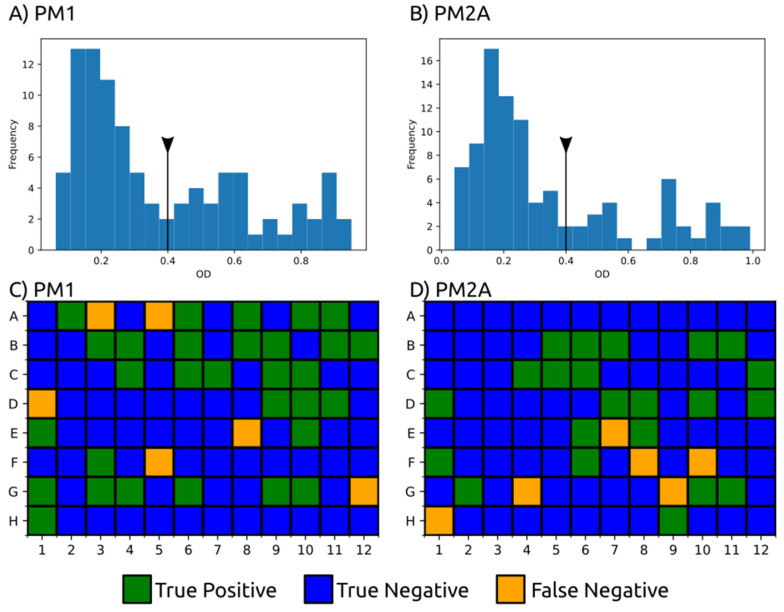
BIOLOG phenotype experiments with carbon sources from PM1 and PM2A. Growth was evaluated by OD 600 after 144 h for PM1 (**A**) and 288 h for PM2A (**B**) with a threshold of 0.4 a.u (black line with triangle), which excludes the normal distribution at low ODs representing no growth. Fifty-two substrates were correctly predicted to growth (true positive, green), and 128 were correctly assigned to nongrowth by iUma22 (true negative, yellow) in plates PM1 (**C**) and PM2A (**D**). Twelve substrates could not be balanced to enable growth in iUma22 (false negative). Results of PM1 and PM2A and an overview of the substrates on the plates are provided as Appendix A.

**Figure 3 jof-08-00524-f003:**
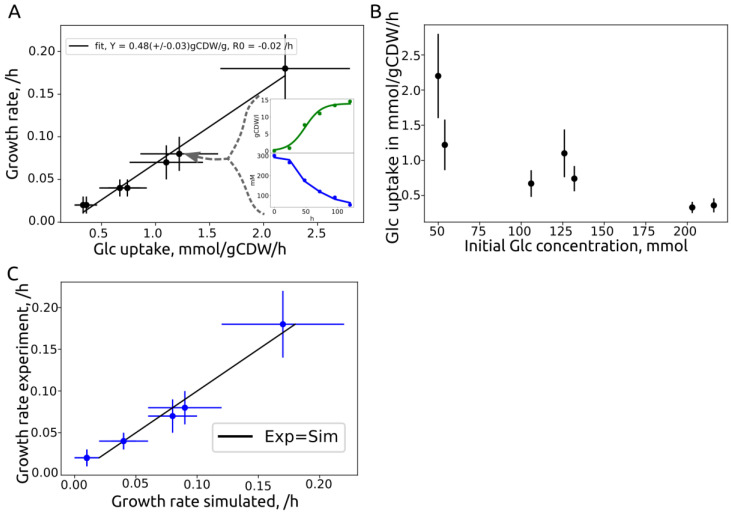
Growth characteristics of *U. maydis* glucose batch cultures from Table 3 and similarity to iUma22 predictions. (**A**) Seven batch experiments on glucose were analyzed to extract growth- and glucose-uptake rates. The linear least-squares correlation provides the biomass yield on glucose with 0.47 +/− 0.03 g_CDW_/g_glc_, and the interception of the x-axis provides the glucose maintenance uptake rate with 0.2 +/− 0.01 mmol/g_CDW_/h. The two inlet figures exemplify the growth rate estimation by a logistic Verhulst equation for growth (green) and linear substrate uptake (blue) for experiment ID ‘50glc’. (**B**) Glucose-uptake rate as a function of the initial glucose level indicating an inverse correlation. (**C**) Simulated and experimental growth rates, with optimal predictions represented by the black line. The individual growth rate data is provided in the Appendix A.

**Figure 4 jof-08-00524-f004:**
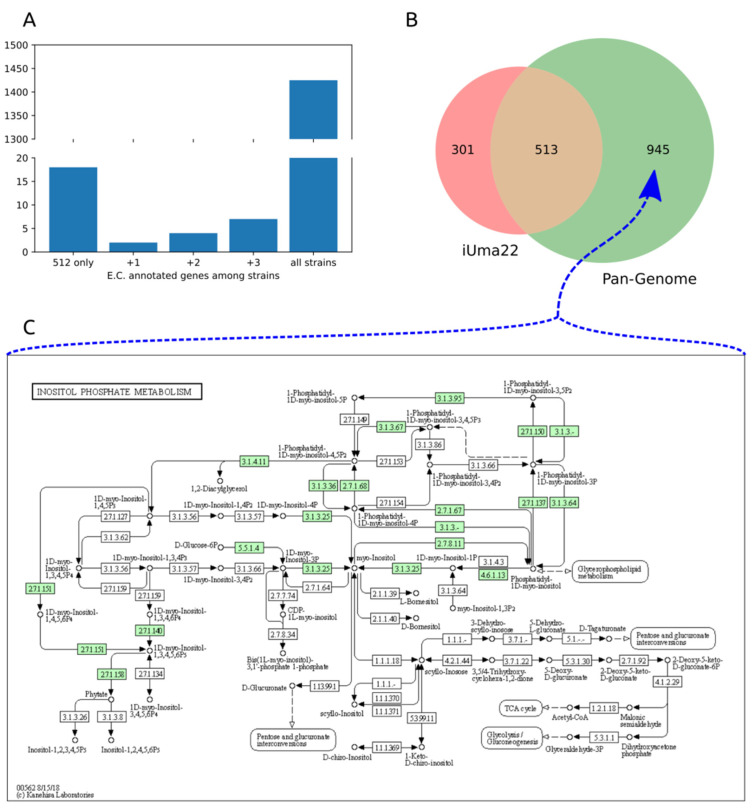
Comparison of enzymes in *U. maydis* strain pangenome and iUma22. (**A**) E.C.-annotated genes in strain 512 that are unique to 512 or shared with the other strains. (**B**) Coverage of the genes in iUma22 of E.C.-annotated genes in the pangenome identified by KAAS. Table 4 show the top five pathways with the most association for iUma22-unique, pangenome-unique, and their intersection. (**C**) The inositol phosphate metabolism contains 20 genes with the highest level of missing genes in iUma22 (Appendix A).

**Table 1 jof-08-00524-t001:** Macromolecular composition of *U. maydis* calculated by linear optimization. The full composition is provided as a Appendix A.

Component	Protein	DNA	RNA	Lipids	Cell Wall
g/100 gCDW	30	0.3	10	40	16

**Table 2 jof-08-00524-t002:** Number of metabolites, reactions, and genes of the genome-scale metabolic model of *U. maydis* iUma22 and, in comparison, the community yeast model (8.5.0 [42]).

Component	iUma22	Yeast 7.6 ^1^
Genes	814	1150
Metabolites	1233	2742
Reactions	1856	4058
Reactions with GPR	1434	2633
Predicted genes ^2^	6909	6464

^1^https://github.com/SysBioChalmers/yeast-GEM, commit 24 June 2021. ^2^https://www.ncbi.nlm.nih.gov/datasets, accessed on 4 February 2022.

**Table 3 jof-08-00524-t003:** Glucose batch growth experiments were performed and used from the literature [4]. The data provided growth and substrate uptake rates for testing iUma22 predictions. Growth results for each experiment is provided in the Appendix A.

Source	ID	Strain	Initial Glc, g/L	Growth Rate, /h	Substrate Rate, mmol/g_CDW_/h	Yield, g_CDW_/g_glc_
This work	2229v1	MB215	50	0.18 +/− 0.04	2.2 +/− 0.6	0.45
Becker et al.	50glc	MB215	54	0.08 +/− 0.02	1.22 +/− 0.36	0.36
This work	130v1	MB215	126	0.07 +/− 0.02	1.1 +/− 0.34	0.33
This work	130v2	MB215	132	0.04 +/− 0.01	0.74 +/− 0.18	0.3
Becker et al.	100glc	MB215	106	0.04 +/− 0.01	0.67 +/− 0.19	0.33
This work	200v1	MB215	203	0.02 +/− 0.01	0.33 +/− 0.08	0.33
This work	200v2	MB215	216	0.02 +/− 0.01	0.55 +/− 0.1	0.33

**Table 4 jof-08-00524-t004:** Top five metabolic pathways with the highest number of missing genes in iUma22 compared to the strain pangenome. The annotation is based on KAAS, considering only KEGG pathways (KAAS outputs for iUma22-unique, -shared, and pangenome-unique provided as Appendix A).

iUma22-Unique	Shared	Pan-Unique
Oxidative phosphor. (42)	Purine (29)	Inositol phosphate (20)
TCA cycle (2)	Pyruvate metabolism (27)	Purine (12)
C5-branched metabolism (1)	Glycolysis (25)	N-Glycan biosynth. (11)
Nitrogen metabolism (1)	Gly, Ser, Thr metab. (24)	(GPI)-anchor biosynth. (11)
Starch and sucrose (1)	Val, Leu, Iso metab. (24)	Starch and sucrose (10)

## Data Availability

The data and the model are available on GitHub: https://github.com/iAMB-RWTH-Aachen/Ustilago_maydis-GEM (accessed on 24 March 2022).

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
