# Peer review of "Ustilago maydis Metabolic Characterization and Growth Quantification with a Genome-Scale Metabolic Model"

_jof, 2022, doi:10.3390/jof8050524_

Round 1
Reviewer 1 Report
Minor points:
- P153: the substrate utilization was conducted using the strain PB1, which was not included in the five strains of pan-genome sequencing and the draft GSMM construction. Any explanation?
- Table 2: please use full word for “Pred.”, to be consistent to other descriptions in the column. Also, “6.909” should be “6,909”?
- Line 134: please provide the full name for cerevisiae, when it appears first time in the text.
Author Response
We are grateful to the feedback from the reviewers and the manuscript improved substantially with their comments! We are also relieved to note, that the reviewers recognize the importance of a genome scale model of Ustilago maydis for the community and the technical soundness of our work.
Response to minor points of Reviewer #1:
- P153: the substrate utilization was conducted using the strain PB1, which was not included in the five strains of pan-genome sequencing and the draft GSMM construction. Any explanation?
Response: The substrate utilization tests are based on the same strain as the model. For the pangenome, we chose different strains for sequencing to get a more diverse overview of the strain differences as the model strain is already sequenced.
- Table 2: please use full word for “Pred.”, to be consistent to other descriptions in the column. Also, “6.909” should be “6,909”?
Response: Corrected
- Line 134: please provide the full name for cerevisiae, when it appears first time in the text.
Response: Corrected
Reviewer 2 Report
The manuscripts presents a metabolic model iUma22 based on genome sequences for the smut fungus U. maydis and the validation for growth on different carbon sources. This smut fungus has been an excellent model organism for decades. Due to its bi-phasic growth, the yeast-stage is of high relevance for biotechnology, and the filamentous form is an excellent model for long-distance transport and infection biology. Therefore, the metabolic model will not only support metabolic engineering of the fungus, but can also help to understand the infection process. Therefore, it is of high relevance to both communities and I will be happy to use it in the future.
However, in the current form, I find the manuscript very hard to read and understand without expertise on modelling. Since Journal of Fungi addresses a broad readership of microbiologist, I recommend to substantially rewrite the paper with a focus on making it more accessible to non-experts in modelling. I have suggested some changes in the detailed points below.
I saw a clear strength in the validation of the model by growth assays, but in my view the aspect should be supported by more data. First, the strains and glucose concentrations in the batch experiments are not consistent (see also comments below) and second I would find it very interesting to also compare the model to growth in the existing hxt1D mutant (Schuler 2015 New Phytologist). This would add a very nice disturbance to the model.
Detailed points:
line 35: please change “…maize harvest but is also used as food itself…” to “…but the tumours are also…”
Line 57-64: This paragraph is very technical with a lot of information on how that model was optimized. I would prefer to have a more general statement about application of the model for research and production.
Line 69 and 75/76: Unfortunately, the genome resources PEDANT and MUMDB are no longer available. Is it possible to give the details where the information could be retrieved today, e.g. ensemble fungi?
Line 158: What is the corresponding OD600 at the start of the experiment?
Figure 2: I would like to see the distribution of the substrates in the plates, so that I do not have to look for them in the Biolog-details. Is it possible to include this as small text in the wells?
Line 210: It would be nice to get more information about these types of plates in the text for readers who are not familiar with the Biolog-System. This would be solved, if the substrates were indicated in figure 2.
Line 213-217: It would be great to explain the contribution of Biolog data to the model in more detail.
Line 224: “ability of the model to grow on proline” sounds strange to me. The model predicts that U. maydis would grow on proline, but the model itself does not grow.
Line 228-9: It is unclear to me where the false positive data occurred and why it is ok to correct them by removing reactions from the model. To make this understandable for microbiologist lacking expertise in modelling, it needs to be explained in more detail.
Chapter 3.3: Glucose uptake and sensing is mentioned in this paragraph. Is it possible to validate the model by measuring the parameters in the existing hxt1-mutants?
Table 3: I miss FB1 at higher initial glucose concentrations and MB215 at 20 g/L for comparison between the two strains. It would be great to mention the genetic differences between the two strains MB215 and FB1 that are known and relevant for carbon utilization.
Chapter 3.4: Is the full genome sequence of MB215 available? Could it be included in the pan-genome?
Author Response
We are grateful to the feedback from the reviewers and the manuscript improved substantially with their comments! We are also relieved to note, that the reviewers recognize the importance of a genome scale model of Ustilago maydis for the community and the technical soundness of our work.
Response to detailed points of Reviewer #2:
line 35: please change “…maize harvest but is also used as food itself…” to “…but the tumours are also…”
Response: Corrected
Line 57-64: This paragraph is very technical with a lot of information on how that model was optimized. I would prefer to have a more general statement about application of the model for research and production.
Response: The text is now more biologically descriptive of the data used for model construction:
Here, we present the first high-quality genome-scale metabolic model for U. maydis called iUma22. Growth phenotype assays based on BIOLOG with 190 substrates were conducted to reveal the metabolic versatility of U. maydis for more realistic model predictions in native habitats. Moreover, growth kinetics across a range of high glucose concentrations was performed that allowed for improved metabolic characterization during biotechnological fermentations. To judge the metabolic completeness of iUma22 as well as metabolic capacities in comparison to the Ustilago genus, a pangenome of annotated enzymes of different U. maydis strains was constructed. The model is freely available via the Biomodels database (MODEL2203250001) and Github (https://github.com/iAMB-RWTH-Aachen/Ustilago_maydis-GEM). The quality of the model was assessed using the Memote evaluation tool, as well as FROG reports for reproducibility.
Line 69 and 75/76: Unfortunately, the genome resources PEDANT and MUMDB are no longer available. Is it possible to give the details where the information could be retrieved today, e.g. ensemble fungi?
Response: We are glad for this remark and now provide links to active databases of sequence and annotation information:
While PEDANT and MUMDB are discontinued, information on sequence and annotation for U. maydis can be accessed via EnsemblFungi (https://fungi.ensembl.org/Ustilago_maydis/Info/Index) [17], MycoCosm (https://mycocosm.jgi.doe.gov/Ustma2_2/Ustma2_2.home.html), NCBI (genome assembly ID: 225285) and Uniprot (Proteome ID: UP000000561).
Line 158: What is the corresponding OD600 at the start of the experiment?
Response: The section 2.4 of the manuscript describes the BIOLOG protocol we followed: ‘The inoculation fluid was prepared with IFY-0 (1.2x), cell suspension and sterile water to obtain a starting turbidity of 62% T, with 100 uL for each well.’ A turbidity of 62% corresponds to ~0.2 optical density according to the converter https://www.pgo-online.com/intl/optical-density-transmission-converter.html.
Figure 2: I would like to see the distribution of the substrates in the plates, so that I do not have to look for them in the Biolog-details. Is it possible to include this as small text in the wells?
Response: With the substrates printed directly into the graph the figure becomes too overloaded. We added a supplementary in which the substrate names are organized according to the plates and thus facilitates the identification of growth patterns.
Line 210: It would be nice to get more information about these types of plates in the text for readers who are not familiar with the Biolog-System. This would be solved, if the substrates were indicated in figure 2.
Response: We added a short explanation of the BIOLOG system in the corresponding Results section and added the BIOLOG plate substrate overview as supplementary.
Line 213-217: It would be great to explain the contribution of Biolog data to the model in more detail.
Response: There are more than 40 reactions added to the model based on positive growth of the Biolog plates. As written, the majority is based on oligosaccaride metabolism and methylated substrates. We have collected the associated reactions in a new Excel file as supplementary for quick reference.
Line 224: “ability of the model to grow on proline” sounds strange to me. The model predicts that U. maydis would grow on proline, but the model itself does not grow.
Response: Agreed! Reformulated as:
The degradation of arginine (PM2A:G4), isoleucine (PM2A:G9) and ornithine (H1) is dys-functional in iUma22, although proline is predicted to enable growth.
Line 228-9: It is unclear to me where the false positive data occurred and why it is ok to correct them by removing reactions from the model. To make this understandable for microbiologist lacking expertise in modelling, it needs to be explained in more detail.
Response: The following explanation is provided:
False positive growth predictions indicated exchange and transport reactions of metabolites in the model that actually could not be imported. Thus, growth predictions of the corresponding metabolites were corrected by removing exchange reactions of the associated metabolites.
Chapter 3.3: Glucose uptake and sensing is mentioned in this paragraph. Is it possible to validate the model by measuring the parameters in the existing hxt1-mutants?
Response: There exists a detailed kinetic characterization of glucose uptake via Hxt1 (Schuler et al., 2015), and it is tempting to include these results to Figure 3B. However, we feel uncomfortable to compare these data directly because the characterization took place in S. cerevisiae by means of heterologous expression and the conversion of the reported rates to uptake units used in the model resulted in values too high for a wild type. The unit conversion from (nmol(Glc)/mgFW/min) to (mmol(Glc)/gCDW/h) itself is empirical with numerous uncertainties. We manually extracted the correlation between cell fresh weight (wet weight) and dry weight (CDW) in S. cerevisiae fermentations as 1 gFW/L ~ 6 gCDW/L from Figure 1 by Aon et al. (2015, doi: 10.1186/s12934-016-0542-3). Note, that the yeast cell densities of Aon et al. are ~10x higher compared to Schuler et al. (OD of 10 and a typical conversion factor of 0.62 gCDW/L/OD, Bionumbers BNID 111182). Overall, in combination with the conversions min->h, nmol->mmol, the conversion is roughly: 1 nmol(Glc)/mgFW/min ~ 10 mmol(Glc)/gCDW/h. Thus, the uptake rate in Schuler et al. at pH=6 amounts to ~30 mmol/gCDW/h, rougly 3x the expected performance of wild type uptake rates (van Dijken et al., 1993, doi: 10.1007/BF00871229).
Table 3: I miss FB1 at higher initial glucose concentrations and MB215 at 20 g/L for comparison between the two strains. It would be great to mention the genetic differences between the two strains MB215 and FB1 that are known and relevant for carbon utilization.
Response: The question of the source of the metabolic transition, as testified by the change in yield, is interesting regardless of whether it is caused by growth rate related processes or by strain specific genetic elements. We have documented both options in text and table, and we feel answering this question is beyond the scope of our manuscript, which is streamlined to provide and document an initial genome scale model for further quantitative studies.
We now realized that Figure 3 was not correctly rendered in the submitted manuscript. Figure 3 is a graphical summary of Table 3. Even more than the table, the Figure indicates the reverse correlation of glucose initial level and uptake rate for even for the MB215 strain alone.
Chapter 3.4: Is the full genome sequence of MB215 available? Could it be included in the pan-genome?
Response: We have not sequenced MB215 and are unaware of a public sequence.
Round 2
Reviewer 2 Report
Thank you very much for integrating my changes. It helps me a lot to understand the manuscript.
There are two points I would like to come back to.
- Chapter 3.3: Glucose uptake and sensing is mentioned in this paragraph. Is it possible to validate the model by measuring the parameters in the existing hxt1-mutants?
Response: There exists a detailed kinetic characterization of glucose uptake via Hxt1 (Schuler et al., 2015), and it is tempting to include these results to Figure 3B. However, we feel uncomfortable to compare these data directly because the characterization took place in S. cerevisiae by means of heterologous expression and the conversion of the reported rates to uptake units used in the model resulted in values too high for a wild type. The unit conversion from (nmol(Glc)/mgFW/min) to (mmol(Glc)/gCDW/h) itself is empirical with numerous uncertainties. We manually extracted the correlation between cell fresh weight (wet weight) and dry weight (CDW) in S. cerevisiae fermentations as 1 gFW/L ~ 6 gCDW/L from Figure 1 by Aon et al. (2015, doi: 10.1186/s12934-016-0542-3). Note, that the yeast cell densities of Aon et al. are ~10x higher compared to Schuler et al. (OD of 10 and a typical conversion factor of 0.62 gCDW/L/OD, Bionumbers BNID 111182). Overall, in combination with the conversions min->h, nmol->mmol, the conversion is roughly: 1 nmol(Glc)/mgFW/min ~ 10 mmol(Glc)/gCDW/h. Thus, the uptake rate in Schuler et al. at pH=6 amounts to ~30 mmol/gCDW/h, rougly 3x the expected performance of wild type uptake rates (van Dijken et al., 1993, doi: 10.1007/BF00871229).
My new comment: I was more thinking of growing the mutant in batch and maybe in the Biolector and comparing the changes to the predicted changes in the model when removing this glucose uptake reaction. This would nicely validate the model by genetic disturbance, but is not an essential point for this manuscript.
- Table 3: I miss FB1 at higher initial glucose concentrations and MB215 at 20 g/L for comparison between the two strains. It would be great to mention the genetic differences between the two strains MB215 and FB1 that are known and relevant for carbon utilization.
Response: The question of the source of the metabolic transition, as testified by the change in yield, is interesting regardless of whether it is caused by growth rate related processes or by strain specific genetic elements. We have documented both options in text and table, and we feel answering this question is beyond the scope of our manuscript, which is streamlined to provide and document an initial genome scale model for further quantitative studies.
We now realized that Figure 3 was not correctly rendered in the submitted manuscript. Figure 3 is a graphical summary of Table 3. Even more than the table, the Figure indicates the reverse correlation of glucose initial level and uptake rate for even for the MB215 strain alone.
My new comment: I agree that the genetic differences are not essential, but I still miss the data for FB1 at higher initial glucose concentrations and for MB215 at 20 g/L. In my view, it is essential to have complete datasets for all concentrations in at least one genetic background. This should be repeated. Alternatively, if the correlation is detectable in the MB215 strain alone, it might be sufficient to simple remove FB1 from the dataset.
Author Response
The growth quantification experiment for strain FB1 was performed in a lower glucose concentration compared to experiments with MB215 and it is impossible to associate the physiologically distinct results to either the strain or the glucose concentration. While this is a very interesting finding, we agree with the reviewer that this result might mislead readers. This observation is not directly linked with the model iUma22 itself, which remains valid without, and thus we follow the recommendation by the reviewer to excise the growth quantification experiment of FB1 from the manuscript. The significance of this observation is given more justice with a dedicated, detailed analysis and a separate report than as a byproduct of a genome scale metabolic reconstruction.
This excision vindicates the remaining comments.